# Study protocol for a population-based observational surveillance study of culture-confirmed neonatal bloodstream infections and meningitis in South Africa: Baby GERMS-SA

Susan Meiring [1,2] Rudzani Mashau,[1] Rindidzani Magobo,[1] Olga Perovic,[1,2] Vanessa Quan,[1] Cheryl Cohen [1,2] Linda de Gouveia,[1] Anne von Gottberg,[1,2] Cheryl Mackay,[3] Mphekwa Thomas Mailula,[4] Rose Phayane,[5] Angela Dramowski,[6] Nelesh P Govender[1,2]

For numbered affiliations see end of article.

**Correspondence to**
Dr Susan Meiring;
SUSAN.MEIRING@NHLS.AC.ZA

## ABSTRACT

**Introduction** Worldwide, neonatal mortality remains high accounting for 47% of childhood deaths in 2019 and including an estimated 500 000 deaths from neonatal infections. While 42% of global neonatal deaths occur in sub-Saharan Africa, there is limited understanding of population-level burden and aetiology of neonatal infections outside tertiary-level institutions.

**Methods and analysis** We aim to implement the first population-level surveillance for bloodstream infections and meningitis among neonates aged <28 days in South Africa. Tier 1 will include national surveillance of culture-confirmed neonatal infections at all public-sector hospitals describing infection incidence risk, pathogen profile and antimicrobial susceptibility by institution, province and healthcare level (2014–2021). Tier 2 (nested within tier 1) will be conducted at six regional neonatal units over 12 months, will compare the clinical characteristics of neonates with early-onset and late-onset infections and identify potentially modifiable risk factors for mortality. Through tier 2, we will determine the antimicrobial susceptibility of neonatal pathogens, evaluate the appropriateness of empiric antibiotic prescribing and determine the genomic epidemiology of multidrug resistant bacterial and fungal pathogens.

**Ethics and dissemination** Ethics clearance was obtained from the Human Research Ethics Committee of the University of the Witwatersrand (M190320). Funding for the study was obtained through a grant from the Bill and Melinda Gates Foundation (OPP1208882). Baby GERMS-SA aims to impact on national policy, resource allocation and neonatal guidelines by describing the national burden of neonatal infections in South Africa. In addition, end-users in neonatal units will benefit from a facility-level dashboard displaying key indicators of the surveillance findings.

### Strengths and limitations of this study

► Baby GERMS-SA will be the first population-level surveillance study to determine the aetiology of culture-confirmed neonatal infections in an African country.

► Two complementary surveillance approaches will be used:
  – Retrospective population-based laboratory surveillance in all public health facilities in South Africa.
  – Prospective enhanced surveillance for clinical data at six neonatal units.

► Providing baseline incidence estimates for culture-confirmed early-onset and late-onset infections in neonatal units will help monitor the impact of future public health interventions aimed at reducing infection-related neonatal mortality.

► The observational nature of the study may underestimate neonatal infection burden in rural districts of South Africa, where specimen taking practices to confirm infections are suboptimal.

childhood deaths from 12.7 million in 1990 to 5.2 million in 2019.[1 2] Neonatal deaths accounted for 47% of all under-5 childhood deaths in 2019, with infectious causes being the third highest contributors to neonatal mortality, following prematurity and intrapartum related events.[2] Infectious diseases caused approximately 500 000 neonatal and 1.5 million under-5 childhood deaths in 2017.[2 3]

While 42% of global neonatal deaths occur in Sub-Saharan Africa, the population-level burden and aetiology of neonatal infections is not well understood.[1 4 5] Studies in Africa have been limited to tertiary-level institutions,

## INTRODUCTION

Worldwide, neonatal mortality remains high, despite a substantial decline in under-5

with no population-based surveillance studies reporting on neonatal infection incidence risks or rates.[5–8] Contributing factors to this lack of data include under-utilisation or unavailability of hospital-based services for neonatal care, suboptimal specimen collection to confirm an infectious disease diagnosis, limited capacity of diagnostic pathology laboratories to detect, identify and characterise neonatal pathogens, absence of appropriate denominator data for calculating incidence risks or rates, lack of clinical data to differentiate between infection types (ie, healthcare-associated infections vs vertical transmission of pathogens causing early-onset sepsis), and limited resources for setting up and maintaining population-based surveillance studies.[9–11]

The South African government seeks to reduce neonatal sepsis rates by 84% nationally by 2025 through various strategies across the continuum of maternal and newborn care.[12 13] However, unless the national burden of laboratory-confirmed neonatal infections occurring at all levels of healthcare in South Africa can be clearly documented, measuring the effectiveness of these interventions against a baseline will be difficult.

We aim to improve the reporting of neonatal infection burden and determine the risk factors for mortality associated with neonatal infections in urban and rural South Africa using a two-tiered surveillance study. We will describe the incidence of culture-confirmed neonatal bloodstream infections and meningitis by province, pathogen-specific aetiology and antimicrobial susceptibility at different levels of healthcare over an 8-year period in tier 1. We will describe the clinical characteristics of culture-confirmed cases, identify modifiable risk factors associated with mortality and describe the antimicrobial susceptibility and genomic epidemiology of multidrug resistant bacterial and fungal pathogens over 12 months in tier 2.

## METHODS

### Hypothesis

We hypothesise that the incidence risk of culture-confirmed neonatal infections has increased over the study period in South Africa, owing to in-hospital transmission of multidrug-resistant organisms. In addition, we hypothesise that neonatal deaths due to infections may be related to modifiable risk factors such as low rates of antenatal steroid use in preterm infants, low rates of breast feeding among neonates who develop infections, and prolonged use of indwelling catheters.[14 15]

### Study objectives

Baby GERMS-SA has three main objectives:
1. To determine the bacterial and fungal aetiology and incidence risk of culture-confirmed infections among neonates presenting to all levels of hospital-based care in South Africa from 2014 to 2021.
2. To confirm the bacterial and fungal aetiology and prevalence of antimicrobial resistance in pathogens

causing neonatal infections at secondary-level healthcare facilities over a 12-month period in South Africa.
3. To determine the characteristics of neonates who are diagnosed with culture-confirmed infections at secondary-level healthcare facilities and identify potentially-modifiable risk factors for death.

### Study design

Two complementary surveillance approaches will be used in this study. First, retrospective population-based surveillance will be established to identify culture-confirmed episodes of neonatal infections occurring in all public health facilities in South Africa from 2014 to 2021, and population denominator data on live births will be used to calculate national and provincial incidence risks of infection (tier 1). Second, prospective enhanced laboratory-based surveillance will be conducted at six sentinel neonatal units to collect detailed clinical data from neonates with infection and to determine risk factors for mortality (tier 2). The overall and pathogen-specific incidence rate of neonatal infections will be calculated at sentinel sites using patient bed-days as a denominator. All-cause mortality rates will also be calculated. The sentinel neonatal units will be selected from a list of secondary level/ regional public-sector hospitals with inpatient neonatal services. Only one institution will be selected per province. In addition, a cross-sectional electronic survey will be conducted at a sample of large public-sector neonatal units in South Africa to determine available bed and staff resources, understand infection prevention and antimicrobial stewardship practices and obtain admission denominator data.

### Definitions

A neonate will be defined as a child aged <28 days with further categorisation into early (0–6 days) and late neonatal periods (7–27 days). The postneonatal period will be defined as the period from 28 to 60 days.

We will use a laboratory-based case definition for neonatal invasive infections based on level 1 of the Brighton Collaboration Neonatal Infections Working Group for neonatal invasive bloodstream infections.[16] This includes any neonate/infant who is admitted to a public-sector hospital with a recognised pathogen (bacteria or fungi) identified using a validated method from a normally-sterile site (blood or cerebrospinal fluid (CSF)) or a normally non-pathogenic organism, for example, coagulase-negative staphylococci isolated from 2 invasive specimen cultures taken at 2 different time points within 14-days. We will use a 14-day period from the date of the first positive culture to define an episode of infection. This case definition makes the assumptions that: (1) the neonate/infant would not have been evaluated for sepsis (ie, had specimens collected for culture) in the absence of clinical signs or a clear clinical indication and (2) the isolated bacteria/ fungi are not contaminants and (3) that most neonatal bacterial or fungal infections can be cleared within 14 days with appropriate

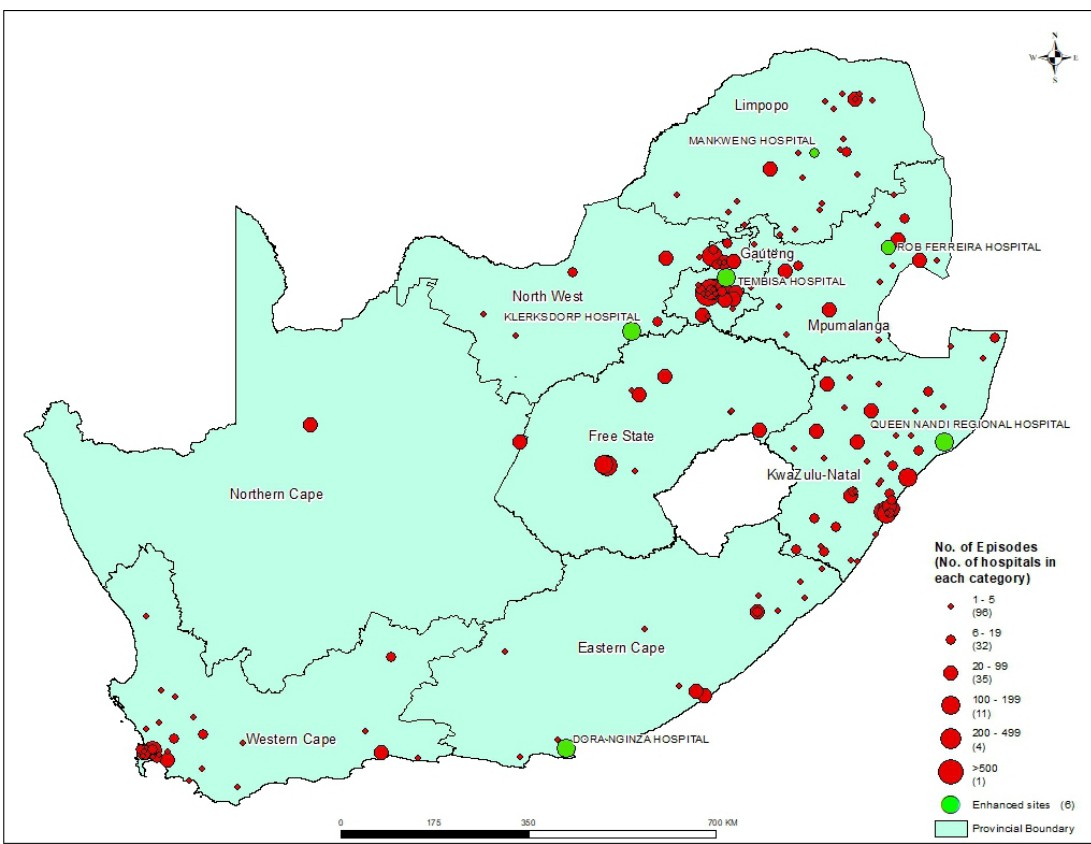

**Figure 1** Map of South Africa showing relative numbers of laboratory-confirmed bacterial and fungal infectious episodes among neonates diagnosed at each Hospital site, 2016 (n=7124). The six tier 2 sentinel surveillance sites (enhanced sites) are indicated in green.

antimicrobial treatment. We will use a 72-hour age cut-off to distinguish early-onset (less than 72 hours since birth) from healthcare-associated (≥72 hours since birth) neonatal infections.

### Study processes, training and analyses
#### Tier 1: national population-based laboratory-based surveillance
Positive blood and CSF microbiology culture results from patients admitted to public health institutions in South Africa will be obtained from a surveillance data warehouse which archives data from TrakCare, the electronic National Health Laboratory Service (NHLS) laboratory information system in use since at least 2014. We will request data on positive blood and CSF cultures among infants aged <12 months for at least an 8-year period (from 1 January 2014 to 31 December 2021). Based on a preliminary analysis of 2016 data, we estimate 7000–8000 laboratory-confirmed neonatal infections to be reported each year from approximately 180 public-sector hospitals (figure 1). The following variables will be requested: laboratory name, province, district, sub-district, hospital name, ward name/type, patient first name and surname, laboratory episode number, data warehouse unique identifier, patient date of birth, date of specimen collection, specimen type, microscopy (including Gram stain and CSF cell counts) and culture result, identification of pathogen and antimicrobial susceptibility results. Patient identifying

information (ie, name, date of birth) will be requested in order to accurately deduplicate records; this is also the current practice in the 'parent' GERMS-SA surveillance programme.[17] This information is essential to distinguish neonatal and maternal specimens (in the first few days of life, clinicians often send specimens labelled with the mother's details). Neonatal date of birth is also an essential piece of information to determine timing of infection (early vs late onset). A national surveillance dataset will be created containing deduplicated laboratory records of neonates with laboratory-confirmed bloodstream and CSF infections. These data will be cleaned and analysed using Stata V.15 (StataCorp).

Data from the first tier of national laboratory-based surveillance will be used to calculate the incidence risk of neonatal sepsis stratified by level of healthcare (district (level 1), regional (level 2) or tertiary/referral (level 3)), geographical region (province, district, subdistrict) and timing of infection. We will use national neonatal unit admissions (if available through the cross-sectional survey) or live births in the total population as denominators for incidence risk calculations.[18] The main analysis (incidence risk calculations) may focus on provinces where specimen-collection practices are more consistent and estimates of incidence risks are more likely to be valid. Missing data will be imputed for the stratified

incidence risk calculations. In line with the National Institute for Communicable Diseases' (NICD) mandate, we will endeavour to make aggregate data publicly available through a neonatal infection dashboard displaying interactive maps and graphs to district level (similar to the antimicrobial resistance dashboard available at www.nicd.ac.za).

### Tier 2: enhanced sentinel site laboratory-confirmed neonatal sepsis surveillance

The six sentinel regional hospitals and their provinces include: Dora Nginza Provincial Hospital (Eastern Cape Province), Tembisa Hospital (Gauteng Province), Mankweng Hospital (Limpopo Province), Klerksdorp (North West Province), Queen Nandi Regional Hospital (KwaZulu-Natal Province) and Rob Ferreira Hospital (Mpumalanga Province) (figure 1). In 2016, 13% (901/7 124) of all culture-confirmed neonatal infection episodes in South Africa occurred in neonates admitted to these facilities (figure 1). NHLS microbiology laboratories serving each of the six sentinel surveillance sites will be requested to prospectively submit any cultured bloodstream or CSF isolate from infants aged <12 months on Dorset transport media (Media Mage, Johannesburg) to the NICD reference laboratories for further characterisation. NICD personnel will provide training to participating laboratories to ensure that neonatal specimens are optimally processed and quality control measures are adhered to. In addition, training on diagnosis of neonatal sepsis and meningitis will be provided to clinicians. Data on basic demographic details of the infants and NHLS laboratory characterisation of the isolates will be transferred from the data warehouse directly into Research Electronic Data Capture (REDCap), an electronic data capture tool hosted at the University of the Witwatersrand, and reference laboratory isolate characterisation and clinical data from the infants will be added to this dataset.[19 20] We will prospectively monitor cases of neonatal infections at the sentinel hospitals and on request, conduct investigations for any potential clusters/outbreaks. These activities are covered by a separate ethics application (WITS HREC reference: M160667—Essential communicable diseases surveillance and outbreak investigation activities of the NICD). For each episode of culture-confirmed infection, we will perform a retrospective medical record review using a standardised data abstraction tool (online supplemental table). Inpatient medical charts will be scanned after patient identifiers are physically blocked by a card with a unique study identifier. These imaged records will be abstracted electronically off-site by trained medical and nursing study personnel into a REDCap database.

A dataset will be generated containing all cases of neonatal infection from the six sentinel hospitals. This will include demographic and baseline clinical details of enrolled participants, clinical updates during their hospital admission, antimicrobial therapy during the admission, in-hospital outcome and detailed laboratory characterisation of the isolate/s causing infection. The following variables will be collected: (1) Isolate data: laboratory name, province, district, sub-district, hospital name, ward name/type, patient name and surname, laboratory episode number, data warehouse unique identifier, date of birth, date of admission, date of specimen collection, specimen type, microscopy and culture result, antimicrobial susceptibility results, and NICD reference laboratory data including whole genome sequencing where applicable; (2) Additional patient data: basic demographic data, type of admitting unit, date of birth, gestational age at birth, mode of delivery, 5 min Apgar score, date of admission, comorbid neonatal conditions, maternal and neonatal risk factors for infection, maternal HIV status (and baby's HIV-PCR result if tested), presence of central venous lines, respiratory support (non-invasive, invasive, surfactant administration), clinical presentation on day of specimen collection, inflammatory markers of infection on day of specimen collection, empiric and directed antibiotic/antifungal therapy (determine if 'appropriate' based on organism susceptibility), complications of infection, in-hospital outcome at 28 days (for neonates) or at the end of admission (for neonates and infants who are admitted for longer periods).

The following analyses will be performed using this dataset: (1) A description of the characteristics of neonates with culture-confirmed bloodstream infection and meningitis by pathogen; (2) A description of the antimicrobial susceptibility of the most important bacterial and fungal pathogens causing neonatal infections in South Africa over time and appropriateness of current empiric regimens for early and late-onset infections (3) The incidence risk of infection by pathogen per 1000 live births in the catchment area or per 1000 inpatient days will be calculated for neonates (0–27 days) (4) An estimation of the outcomes following neonatal infection and the risk factors associated with death (5) Multivariable analyses will be performed to determine potentially modifiable risk factors for mortality for various subgroups (eg, effect of receiving antenatal steroids on mortality among preterm infants; effect of feeding mode on mortality; effect of prolonged use of indwelling catheters on mortality among those with late-onset healthcare-associated infections). For each analysis, we will adjust for potential confounders (such as sex, birth weight, preterm birth) as appropriate.

As this dataset will contain confidential information whereby individuals could potentially be identified, the complete dataset will not be available publically. However, should external researchers request any of this information, completely deidentified data may be released following signing of a data-sharing agreement between the NICD and the requestor.

### Electronic survey of neonatal units with previously identified episodes of neonatal sepsis

A baseline survey of neonatal units will be conducted in 2020 to improve our understanding of the current functioning of neonatal units at various healthcare levels in

South Africa. A standardised questionnaire will be sent to all dedicated in-patient neonatal units for the facility manager to complete in hard copy, electronically or online using the SurveyMonkey application. Of 7 124 episodes of laboratory-confirmed neonatal infections diagnosed at 179 South African public hospitals in 2016, 97% (6919 episodes) were from 90 hospitals that had >5 episodes of infection. We will approach these 90 hospitals for the neonatal unit survey (figure 1). Variables to be collected will include: hospital location, bed census in the neonatal unit, number of neonatal unit staff members, on-site or off-site laboratory services, infection prevention and control (IPC) and clinical microbiologist support for the unit and detailed statistics on number of patient-days per month and deaths per month from the neonatal unit in 2019/2020. Clinical criteria used for suspicion of sepsis or bloodstream infections, criteria for obtaining blood/CSF specimens for culture from neonates and institutional antibiotic guidance for early-onset and late-onset sepsis will be also be collected. Completed questionnaire data will be captured into Microsoft Excel and analysis performed using Stata. We will use denominator data obtained through the survey (eg, admissions, patient-days) for incidence risk/ rate calculations at tier 2 facilities.

## Data dissemination

The information gained through the surveillance study will be shared with the South African Department of Health, and various other in-country and international stakeholders. Internally, the data will be presented to the National Advisory Group on Immunisations, the Ministerial Advisory Committee on Antimicrobial Resistance, Neonatal Sepsis Task Force and the multisectoral National Outbreak Response Team, as well as at local and international conferences.[21] We plan to publish the results in a policy briefing and in relevant peer-reviewed medical journals. A facility-level dashboard used to display key indicators based on the surveillance data will be set up and made available to end-users in neonatal units.

## Beneficiaries

We will gain a better understanding of the burden of neonatal infections and the antimicrobial susceptibility and molecular relatedness of neonatal bacterial and fungal pathogens in an upper-middle-income country, particularly in secondary-level institutions serving periurban and rural communities. The National Department of Health could use these data to design appropriate interventions such as antimicrobial stewardship and IPC programmes, to prioritise facilities requiring urgent intervention and to tailor these interventions for those at highest risk of neonatal sepsis. The hospitals at which we will conduct enhanced surveillance will benefit from the additional information that will come from further characterisation of the isolates causing neonatal infections and gain an understanding on how they can tailor their empiric antimicrobial regimens to fit the spectrum of organisms that are being cultured. Individual neonates at these hospitals will thus benefit from the doctors adjusting their empirical therapy accordingly. We will strongly encourage enhanced surveillance sites to implement local antimicrobial stewardship and IPC programmes for neonatal units and will design facility-level reports based on their local surveillance data to allow them to monitor key indicators for neonatal sepsis. Policy-makers can use the data on burden of disease, mortality and risk factors associated with neonatal sepsis and the aetiological patterns of pathogens causing neonatal sepsis to align their strategies on the Continuum of Maternal and Newborn Care to help meet South Africa's goal to reduce neonatal sepsis by 84% by the year 2025. We hope that by setting up this surveillance study, we will facilitate future sustainable funding of the project and will be able to objectively record the change in incidence risk of neonatal infections over time as new interventions are implemented. Ultimately, we hope that policies put in place through the data generated by this project will save the lives of many newborn babies and improve the quality of life of others in the years ahead.

## Patient and public involvement

The study design and protocol was discussed with paediatricians and neonatologists at various institutions across South Africa, as well as at the launch of the Neonatal Sepsis Task Force at the United SA Neonatal Association conference in Port Elizabeth, South Africa, on 13 September 2019.

## Ethics and funding

The study has been approved by the Human Research Ethics Committee of the University of the Witwatersrand (M190320). Approvals for the tier 2 surveillance study were received from each provincial research committee through registration on the National Health Research Database. Funding for the study was awarded as a grant to NG from the Bill & Melinda Gates Foundation (OPP1208882).

## DISCUSSION

Unless baseline data on neonatal infections are reliably and systematically collected and analysed, there will be no objective record against which interventions aimed at reducing burden of disease in this vulnerable population can be measured. The Baby GERMS-SA national surveillance system will provide a robust platform to determine national incidence risk of neonatal infection by pathogen, level of hospital care and geographical region. This surveillance study will also be used to assess trends in the incidence risk of neonatal infections over time, thus providing an objective record by which to measure the impact of any intervention implemented in future.

The enhanced surveillance tier will focus on neonatal sepsis and meningitis occurring at sentinel hospitals. This surveillance tier will provide insight into the similarities/

differences in pathogens and their antimicrobial susceptibility patterns causing neonatal sepsis by level of healthcare, and risk factors for mortality among neonates admitted to regional hospitals.

The NICD has experience in conducting surveillance for laboratory-confirmed meningitis as well as active surveillance of several invasive bacterial and fungal diseases at a national level.[22 23] We have previously published data indicating how public health interventions such as new vaccines, antiretroviral treatment, cryptococcal antigen screening and treatment have reduced the burden of disease/mortality caused by *Streptococcus pneumoniae, Haemophilus influenzae* and *Cryptococcus neoformans* in South Africa.[24–26] We realise the value of having robust data prior to implementation of public health interventions in reporting the effectiveness of these interventions over time. We also conduct surveillance on selected healthcare-associated bloodstream infections and understand the variation of antimicrobial resistance and pathogen profiles by healthcare level and province.[27] We are aware that the study findings rely on adequate specimen collection and laboratory diagnostic capacity of each neonatal unit. Specimen taking practices might differ between units and therefore the surveillance may underestimate neonatal infection burden, especially in rural districts of South Africa. Tier 1's comprehensive dataset should be able to provide an expected culture-positivity rate to describe the extent of this phenomenon. A major strength of our project is that we will gather complete data on laboratory-confirmed bloodstream infections and meningitis from the entire public-sector population and in-depth data from selected secondary-level sites in six provinces.

Ultimately, these surveillance data can be used to address Sustainable Development Goal 3 by aiming to improve neonatal and child health by using a two-tiered laboratory-based surveillance programme to gain a deeper understanding of the aetiology and burden of neonatal sepsis, with a future aim of addressing these factors and thus reducing neonatal morbidity and mortality in low-income and middle-income settings.

**Author affiliations**
[1]National Institute for Communicable Diseases, a division of the National Health Laboratory Service, Johannesburg, South Africa
[2]University of the Witwatersrand, Faculty of Health Sciences, Johannesburg, South Africa
[3]Department of Paediatrics and Child Health, Dora Nginza Hospital, Port Elizabeth, South Africa
[4]Department of Paediatrics and Child Health, Mankweng Regional Hospital Mankweng, Mankweng, South Africa
[5]Department of Paediatrics and Child Health, Tembisa Provincial Hospital, Johannesburg, South Africa
[6]Department of Paediactrics and Child Health, Division of Paediatric Infectious Diseases, Faculty of Medicine and Health Sciences, Stellenbosch University, Stellenbosch, South Africa

**Acknowledgements** The authors would like to acknowledge the following clinical and laboratory staff involved in the protocol design, facilitation and collection of data for the Baby GERMS-SA project: Dora Nginza Hospital: Phunyezwa Mzayiya (laboratory), Shareef Abrahams (pathologist), Vanessa Pearce (laboratory), Zikhona Gabazana (research assistant (RA)), Melissa Ngubane (RA), Badikazi Matiwana (RA); Klerksdorp Hospital: Omphile Mekgoe (clinician), Sebabatso Khantsi (laboratory), Bernard Motsetse (RA), LouisaPhalatse (RA); Mankweng Hospital: Ruth Lekalakala (pathologist), Tebogo Modiba (RA), Molly Morapeli (RA); National Institute for Communicable Diseases: Linda Erasmus (pathologist), Danie Erwee (clinician), Juliet Paxton (clinician), Siyanda Dlamini (laboratory), Marshagne Smith (laboratory), Ruth Mpembe (laboratory), Ntombi Dube (administrator), Relebohile Ramatsa (RA), Thembekile Zwane (masters student), Sibongile Walaza (medical epidemiologist), Erika van Schalkwyk (medical epidemiologist); Queen Nandi Hospital: ; Constance Kapongo (clinician), Meluleki Mthimkhulu (laboratory), Sandra Maphumulo (pathologist), Dianette Pearce (RA); Rob Ferreira Hospital: Lerato Motjale (clinician), Thulisile Maphosa (clinician), Greta Hoyland (laboratory), Sindile Ntuli (pathologist), Lesley Ingle (RA); Tembisa Hospital: Harishia Naidoo (clinician), Ramatlhwa Kekana (laboratory), Dina Pombo (laboratory). The authors would like to acknowledge Jimmy Khosa from the NICD for the production of the ARCGIS map for figure 1.

**Contributors** The authors contributed in the following ways: Substantial contribution to conception and design of the study: SM, RMat, RMag, OP, VQ, CC, LdG, AvG, CM, MTM, RP, AD and NG. Revising the manuscript for important intellectual content: SM, RMat, NG, AD, VQ and OP. Final approval of manuscript: SM, RMat, RMag, OP, VQ, CC, LdG, AvG, CM, MTM, RP, AD and NG. Guarantor: SM is the guarantor of this publication and takes full responsibility for the content, the decision to publish and the completed work.

**Funding** Funding for the study was awarded as a grant to NG from the Bill & Melinda Gates Foundation (OPP1208882).

**Map disclaimer** The inclusion of any map (including the depiction of any boundaries therein), or of any geographic or locational reference, does not imply the expression of any opinion whatsoever on the part of BMJ concerning the legal status of any country, territory, jurisdiction or area or of its authorities. Any such expression remains solely that of the relevant source and is not endorsed by BMJ. Maps are provided without any warranty of any kind, either express or implied.

**Competing interests** None declared.

**Patient consent for publication** Not applicable.

**Provenance and peer review** Not commissioned; externally peer reviewed.

**ORCID iDs**
Susan Meiring http://orcid.org/0000-0003-4508-5469
Cheryl Cohen http://orcid.org/0000-0003-0376-2302

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
