## [Reviewer comments · BMJ Open]

ARTICLE DETAILS

TITLE (PROVISIONAL)	A study protocol for a population-based observational surveillance study of culture-confirmed neonatal bloodstream infections and meningitis in South Africa: Baby GERMS-SA
AUTHORS	Meiring, Susan; Mathebula, Rudzani; Magobo, Rindidzani; Perovic, Olga; Quan, Vanessa; Cohen, Cheryl; de Gouveia, Linda; von Gottberg, Anne; Mackay, Cheryl; Mailula, Mphekwa; Phayane, Rose; Dramowski, Angela; Govender, Nelesh

VERSION 1 – REVIEW

REVIEWER	Kurtzhals, Jørgen University of Copenhagen, ISIM
REVIEW RETURNED	23-Mar-2021

GENERAL COMMENTS	This is not a novel study design but a highly important survey including plans for longterm, continuous monitoring of neonatal infections in South Africa. The authors adequately address concerns about incomplete coverage with blood cultures in peripheral health facilities and the underreporting that this will cause. It should be noted that causality cannot be inferred from this kind of study, and the modifiable risk factors should ideally be confirmed in subsequent intervention studies and clinical trials. However, this surveillance system would allow for time series studies of e.g. policy changes.
---

VERSION 1 – AUTHOR RESPONSE

Reviewer: 1

Dr. Jørgen Kurtzhals, University of Copenhagen

Comments to the Author:

This is not a novel study design but a highly important survey including plans for long-term, continuous monitoring of neonatal infections in South Africa. The authors adequately address concerns about incomplete coverage with blood cultures in peripheral health facilities and the underreporting that this will cause. It should be noted that causality cannot be inferred from this kind of study, and the modifiable risk factors should ideally be confirmed in subsequent intervention studies and clinical trials. However, this surveillance system would allow for time series studies of e.g. policy changes.

Thank you for your time to review the manuscript. We appreciate your concerns about not being able to confer causality using surveillance data and will be mindful of this when analysing the study results.

Reviewer: 1

Competing interests of Reviewer: None